



Spatial variations in zooplankton community structure along the Japanese coastline in the Japan Sea: influence of the coastal current

Taketoshi Kodama[1, *], Taku Wagawa[1], Naoki Iguchi[1], Yoshitake Takada[1], Takashi Takahashi[2], Ken-Ichi Fukudome[1], Haruyuki Morimoto[1], Tsuneo Goto[1]

[1]Japan Sea National Fisheries Research Institute, Japan Fisheries Research and Education Agency, Niigata, 951-8121, Japan
[2]Joetsu Environmental Science Center, Joetsu, 942-0063, Japan
*Correspondence to*: Taketoshi Kodama (takekodama@affrc.go.jp)

**Abstract.** This study evaluates spatial variations in zooplankton community structure and potential controlling factors along the Japanese coast under the influence of the coastal branch of the Tsushima Warm Current (CBTWC). Variations in the density of morphologically-identified zooplankton in the surface layer in May were investigated for a 15-year period. The density of zooplankton (individuals per cubic meter) varied between sampling stations, but there was no consistent west–east trend. Instead, there were different zooplankton community structures in the west and east, with that in Toyama Bay particularly distinct: *Corycaeus affinis* and *Calanus sinicus* were dominant in the west and *Oithona atlantica* was dominant in Toyama Bay. Distance-based redundancy analysis (db-RDA) was used to characterize the variation in zooplankton community structure, and four axes (RD1–4) provided significant explanation. RD2–4 only explained <6.5% of variation in the zooplankton community and did not show significant spatial difference; however, RD1, which explained 84.6% of variation, did vary spatially. Positive and negative species scores on RD1 represent warm- and cold-water species, respectively, and their variation was mainly explained by water column mean temperature. Water column mean temperature is highly correlated with mean and maximum salinity, and it is considered to vary spatially with the CBTWC. The CBTWC intrusion to cold Toyama Bay is weak and occasional due to the submarine canyon structure of the bay. Therefore, the varying bathymetric characteristics along the Japanese coast of the Japan Sea generate the spatial variation in zooplankton community structure, and dominance of warm-water species can be considered an indicator of the CBTWC.

## 1 Introduction

Ocean currents transport water and climate properties and distributions of dissolved and particulate matter (Thorpe, 2010). The biological importance of ocean currents has been accepted for a century, since Hjort (1914), and is especially applicable in coastal areas which only cover 7% of the ocean but support 90% of global fish catches (Pauly et al., 2002). Plankton community structure varies in accordance with currents (Russell, 1935), and changes in ocean currents cause fisheries regimes to shift, such as from sardine to anchovy and back (Chavez et al., 2003).

In the Japan Sea (Sea of Japan), which is a marginal sea of the western North Pacific, the Tsushima Warm Current (TWC) flows from west (connected to the East China Sea at the Tsushima Strait) to east (connected to the western North Pacific and the Okhotsk Sea at the Tsugaru and Soya straits, respectively). The TWC has three branches (Kawabe, 1982): the coastal (first) branch (CBTWC) flows along the Japanese coast following the shelf edge; the second branch flows on the offshore side of the first branch; the third branch (also known as East Korean Warm Current) flows along the Korean Peninsula up to ~39°N. As well as the other ocean currents, the TWC transports water properties such as heat, freshwater, nutrients, and phytoplankton and affects climate and marine systems in the Japan Sea and the Japanese Archipelago (Isobe et al., 2002; Hirose et al., 1996;





Hirose et al., 2009; Onitsuka et al., 2010; Kodama et al., 2016; Kodama et al., 2015), and is the spawn and nursery field of important fisheries resources (Goto, 1998, 2002; Ohshimo et al., 2017; Kanaji et al., 2009)

Zooplankton are transported by the TWC: the distributions of the giant jellyfish, *Nemopilema nomurai*, which originates from the Chinese coastal area of the East China Sea (Kitajima et al., 2015; Uye, 2008), and the giant salps, *Thetys vagina* (Iguchi and Kidokoro, 2006), depend on the TWC in the Japan Sea. However, studies of the relationship between the TWC and small zooplankton such as copepods are very limited, especially concerning their spatial distribution. In the TWC, zooplankton biomass varies interdecadally (Hirota and Hasegawa, 1999; Kang et al., 2002; Chiba et al., 2005), and community structure varies seasonally in accordance with water temperature, and interannually with the thickness of the TWC (Chiba and Saino, 2003); however, the advection of small zooplankton and its implication for communities have been not discussed, except for studies on tintinnid distributions in the Tsushima Strait (Kim et al., 2012).

Zooplankton biomass is homogeneous in the CBTWC area, along the Japanese coastline (Hirota and Hasegawa, 1999), but from two previous studies, Iguchi and Tsujimoto (1997) and Iguchi et al. (1999), it can be identified heterogeneity in zooplankton community structure during spring bloom period: warm-water species (*Calanus sinicus*, *Corycaeus affinis*, and *Palacalanus pavrus*) are dominant in Wakasa Bay, in the western part of the Japan Sea, while cold-water species (*Pseudocalanus newmani*, *Metridia pacifica*, and *Oithona atlantica*) are dominant in Toyama Bay, in the east. Although these observations were conducted in only one year and may not be representative of longer term trends in zooplankton community structure, the question of why this spatial difference of zooplankton community structure might occur in the western and eastern parts of the Japan Sea along the Japanese coast is important. Hence, in this study, we aim to provide a better understanding of the spatial variations in zooplankton community structure in spring along the Japanese coast in the Japan Sea and their causal mechanisms. In particular, the influence of the CBTWC is evaluated using a 15-year data set and multivariate analysis, which is useful for explaining the heterogeneity of zooplankton community structure (Field et al., 1982).

## 2. Materials and methods

### 2.1. On-board observations

On-board observations were conducted in May from 1999–2013, by T/V *Mizunagi* of Kyoto Prefecture (from 1999 to 2011) and R/V *Mizuho-maru* of the Japan Sea National Fisheries Research Institute (after 2012), at 26 stations (Stn. 1–26) set along the Japanese coast from Toyama Bay to Wakasa Bay (Fig. 1). Observations were conducted at each cruise for one week (Fig. 2a), in the middle of May until 2010, and at the end of May after 2011. The stations are located near the Japanese coast (<40 km from Honshu) and close to the continental shelf or the shelf edge, with water depths of 40–1230 m. The presence of a submarine canyon structure in Toyama Bay accounts for the relatively deep bottom depths of Stns. 3, 4, 5, and 8 (990, 700, 330, and 1230 m, respectively, Fig. 1). Additionally, Stns. 3–5 are in river mouths, and Stn. 7 is located in the mouth of Nanao Bay, which has a mean bottom depth of <20 m.

At each station, vertical profiles of temperature and salinity were recorded using an STD (Salinity-Temperature-Depth) sensor (Alec Electronics, AST1000) during the *Mizunagi* cruises (until 2011) or a CTD (Conductivity-Temperature-Depth) sensor (Seabird, SBE9plus) during the *Mizuho-maru* cruises (after 2012) from the surface to 150 m depth or 5 m above the bottom (but only 5 m pitch during 1999–2001). Temperature and salinity were omitted at two stations, Stn. 4 in 1999 and Stn. 14 in





2001, due to mechanical malfunction. As the temperature- and density-based depth of the mixed layer ($Z_m$) is similar in the Japan Sea (Lim et al., 2012), $Z_m$ was defined as that at which the temperature is 0.5ºC below the temperature at 5 m depth, as modified from Levitus (1982).

Zooplankton were collected with vertical hauls of a long Norpac net (Nytal 52GG, 335 μm mesh, 0.45 m mouth diameter,

Rigo) from 150 m depth or 5 m above the bottom to the surface. A Rigo flowmeter, calibrated before and after every cruise, was installed at the mouth of the net for estimation of filtered water volume. Since macrozooplankton generally show diel vertical migration, our observations were mainly conducted during daytime (from 6 AM to 7 PM); with observations at six stations conducted after 7 PM (Fig. 2b). Zooplankton samples were fixed soon after sampling with neutral formalin and stored at room temperature until morphological identification. Individuals were identified to species level based on Chihara and

Murano (1997), and then the density of each zooplankton species was calculated (individuals per cubic meter, inds. m$^{-3}$). In 1999, samples were lost at Stns. 10 and 13 due to decay.

### 2.2. Other hydrographic data

The temperature of the surface water layer was obtained at 11 fixed points near the zooplankton sampling stations between 2002–2009, from Toyama Bay to Wasaka Bay including Nanao Bay, using data from the Japan Oceanographic Data Center

(http://www.jodc.go.jp, Fig. 1). Temperature sensors were set at a depth of 2–15 m. Temperature was recorded approximately daily (sometimes weekends were excluded) and the 5-day mean temperature calculated.

Daily sea surface absolute height (SSH) and absolute current velocity data were derived from merged satellite data provided by AVISO (http://aviso.altimetry.fr) between 1999–2013 at a spatial resolution of 0.25º × 0.25º.

No primary productivity parameters were measured during the *Mizunagi* cruises, and thus, the monthly composite sea surface

chlorophyll *a* (SSChl *a*) concentration was derived from GlobColour (http://hermes.acri.fr), which uses merged sensor GSM (Garver-Siegel-Maritoren) products (Maritorena et al., 2010). The spatial resolution was 4 × 4 km, and the SSChl *a* concentration of a station was taken from the nearest neighbor data point. The validation of daily SSChl *a* data in Toyama Bay has already been reported by Terauchi et al. (2014b). The monthly composite data was chosen in our study to avoid having many data blanks.

Sea surface heat flux (Qnet) was calculated by using data sets derived from the Japanese 55-year Reanalysis (http://jra.kishou.go.jp), with a spatial resolution of approximately 0.5º × 0.5º, and the temporal resolution was set as monthly in this study. Qnet was calculated using the following equation;

Qnet = Qdswr + Qdlwr – (Quswr + Qulwr + Qlh + Qsh)         …(1)

In (1), Qdswr, Qdlwr, Quswr, Qulwr, Qlh, and Qsh represent downward short-wave radiation, downward long-wave radiation,

upward short-wave radiation, upward long-wave radiation, latent heat flux, and sensible heat flux, respectively.

### 2.3. Statistical analysis

Hypothesis-driven canonical analysis was used to evaluate the zooplankton community and environmental influences on zooplankton community structure. The calculation was performed using R software (R Core Team, 2017). Canonical analysis is a combination of an ordination technique and regression analysis. Four approaches are applicable for reduced space

ordination applied to zooplankton community structures: principal component analysis (PCA), correspondence analysis (CA),





nonmetric multidimensional scaling (NMDS), and principal coordinate analysis (PCoA) (Ramette, 2007). PCA is most frequently used in exploratory analysis as well as for clustering (Ramette, 2007), however, there are some prerequisites: the data must be quantitative, have a small number of blanks (null data), and show multivariate normality (Legendre and Legendre, 2012). Our plankton community data was quantitative, and to avoid too many blanks (null data), populations were excluded

from the analysis if they met any of the following rules: 1) never above 5% of total numerical abundance in all samples; 2) absent at all 26 stations in one-third of the observation periods (five years); and 3) not able to be identified to species level. However, *Mardia*'s test in MVN package (Korkmaz et al., 2014) demonstrated our community data were not multivariate normal ($p < 0.001$), therefore, PCoA was applied. The NMDS approach was also appropriate, but for comparison of the environmental parameters using canonical analysis, PCoA was considered to be superior.

For hypothesis-driven canonical analysis, we used distance-base redundancy analysis (db-RDA), with PCoA as the ordination technique (Ramette, 2007). Before the db-RDA, multicollinearity in the environmental factors was checked using the variance inflation factor (VIF). Usually, VIF values of 10 are taken as the threshold of multicollinearity (Borcard et al., 2011), but in this study, the threshold was applied more strictly: only parameters with a VIF <3 were selected. The environmental parameters are listed in Table 1: temperature at 5 m depth (SST), water column mean temperature (from 0 to 150 m or bottom-5 m depth;

mean T), minimum temperature (corresponding to temperature at 150 m or bottom-5 m depth; mini T), salinity at a 5 m (SSS), water column mean salinity (mean S), maximum salinity (max S), $Z_m$, and SSChl $a$ as the water quality; bottom depth as the geographical information; and day of May, haul depth, and observation time of day as the station information. Initially, the VIF values of mean T, SST, mini T, mean S, max S, day of May, and haul depth were >3 (Table 1); this means they exhibited similar spatio-temporal variations. We used only mean T for the db-RDA, and discarded the other high VIF parameters. As

a result, VIF values of the remaining parameters were <2.0 (Table 1). We then checked that the parameters significantly explained the zooplankton community variations based on AIC (Akaike information criterion) (Blanchet et al., 2008); as a result, time of sampling was excluded from the analysis. Therefore, the remaining environmental parameters comprised mean T, SSS, $Z_m$, SSChl $a$, and bottom depth. Year is possible explanatory parameter, but as this study focused on spatial variations, the effect of year was removed using partial db-RDA.

After the remaining environmental factors were standardized to Z-scores, scaling to a mean of 0 and standard deviation of 1, db-RDA (including preprocessing) was conducted using the VEGAN package (Oksanen et al., 2007), and detailed codes were based on Borcard et al. (2011). The distance between species was calculated by the Bray method. The db-RDA outputs scores for environmental parameters, stations, and species. The contribution of environmental factors to the axes was evaluated in terms of the coefficient values of the explanatory equations.

**3. Results**

**3.1. Environmental conditions**

Mean T varied from 10.96–17.28 ℃ and showed a decreasing tendency from west–east along the coastline (Fig. 2c). The 15-years mean T was 16°C at Stn. 26 and lowered to <12°C at Stns. 1–2, and the west–east decrease in mean T was reproduced in every year. The spatial trends in mini T, SST, mean S, and max S were similar to mean T (Fig. 2d–g), as shown by the high

VIF values (Table 1). Mini T was also affected by the bottom/haul depth (Fig. 2d). SSS ranged from 29.0–34.58; lower values



(<33) were often observed at Stns. 1–9, which are located in Toyama Bay where large rivers flow (Fig. 2h). $Z_m$ ranged from 5–118 m (Fig. 2i) and did not show significant spatial variation (ANOVA, $p = 0.78$). The greatest $Z_m$ was observed in 2007, which is shown as outliers in Fig. 2i. Monthly SSChl $a$ concentration was variable (Fig. 2j), with high concentrations (>2 mg m$^{-3}$) only observed at stations in Wakasa and Toyama bays (Fig. 2h).

The 5-day mean temperature at the eight fixed points between the northern part of Noto Peninsula and Wakasa Bay was lowest at the end of February or beginning of March, while it was one month later in Toyama Bay (Fig. 3). All areas showed similar May temperatures, varying by only 0.10–0.15℃, which differed from our observations. Temporal variations in the 5-day temperature for Nanao Bay were like those in the areas between Noto Peninsula and Wakasa Bay (Fig. 3).

SSH was lower in Toyama Bay than any of the other areas, with an area of high SSH extending from the Noto Peninsula to

the north–east (Fig. 4a), and current velocity was <15 cm s$^{-1}$ (Fig. 4b). The monthly mean sea surface heat flux changed from negative to positive in April (Fig. 5a), and the mean sea surface heat flux in March–May showed no spatial differences (Fig. 5b).

### 3.2. Zooplankton abundance and community structures

Overall, 78 of the 388 samples were identified to species level and 25 groups to genus level; in specimens unidentified to

species level, key parts for identification were destroyed. Total abundance largely varied from 48–2933 inds. m$^{-3}$ (mean ± SD: 649 ± 418 inds. m$^{-3}$); there was no clear west–east tendency in abundance (Fig. 6a). The highest mean abundances were recorded at Stns. 3 and 7 in Toyama Bay, with 1009 ± 613 and 975 ± 800 inds. m$^{-3}$, respectively, and Stn. 24 in Wakasa Bay recorded the lowest with 361 ± 125 inds. m$^{-3}$. There was no trend in the temporal variation of total abundance (Fig. 6b).

The top 10 dominant species from an average of all stations contained seven copepods (*Corycaeus affinis*, *Oithona atlantica*,

*Calanus sinicus*, *Ctenocalanus vanus*, *Paracalanus parvus* s.l., *Oithona plumifera,* and *Pseudocalanus newmani*), plus one each of Appendicularia (*Oikopleura longicauda*), Euphausiacea (*Euphasia pacifica*), and Cladocera (*Evadne nordmanni*). Spatially, the dominant species differed in the west and east: in the west, *C. affinis* and *C. sinicus* were the first and second most dominant species on average, while *O. atlantica* was most dominant in the east, apart from Stn. 7 (Fig. 6c). *C. affinis* and *E. nordmanni* showed the largest temporal variation (Fig. 6d): *C. affinis* was abundant in 2006 and 2010, and *E. nordmanni*

was dominant in 2004.

Spatial variations in the abundance of each species are shown in Fig. 7. The abundance of *C. affinis* and *C. sinicus* varied irregularly, but was higher in the west: the median for *C. affinis* peaked at Stn. 16, located in the western part of Noto Peninsula. *C. vanus*, *O. plumifera,* and *O. longicauda* also showed significantly higher abundance in the west, while *E. pacifica*, *P. parvus* s.l., *P. newmani,* and *O. atlantica* were higher in the east. *E. nordmanni* did not show a west–east gradient; its abundance was

high at Stns. 7, 9, 10, and 17, but temporal variations were dominant.

### 3.3 Multivariate analysis

After preprocessing, 25 zooplankton species remained for multivariate analysis. In addition to the top 10 species, there were two Chaetognatha (*Sagitta minima* and *S. nagae*), two Cladocera (*Podon leuckarti* and *Penilia avirostris*), nine copepods (*Metridia pacifica*, *Acartia omorii*, *Candacia bipinnata*, *Centropages bradyi*, *Mesocalanus tenuicornis*, *Neocalanus*

*plumchrus*, *Clausocalanus pergens*, *Oncaea mediterranea,* and *Oncaea venusta)*, one Hyperiidea (*Themisto japonica*) and one



Appendicularia (*Fritillaria pellucida*). The 25 species accounted for 94% of total abundance on average, although only ~70% at some stations. At the latter stations, *Oithona longispina* and/or *Oithona* spp. were highly abundant after 2010.

The db-RDA results for total, conditioned, and constrained inertial were 112.63, 1.19, and 17.95, respectively. Four axes of zooplankton community structure variation were significantly explained in the db-RDA: the eigenvalues of the first, second,

third, and fourth axes (RD1–4, respectively) explained 84.6, 6.5, 4.7, and 3.1% of zooplankton community structure variability, respectively.

The relationship between species and environmental parameters is shown in Fig. 8, after scaling the site score using biplot diagrams (Fig. 8); the site score was removed from Fig. 8 since their plot was invisible due to their numbers ($n = 388$). First, on the combination of RD1 and RD2, mean T and SSS plot as neighbors, and bottom depth is opposite: the scores of mean T

and SSS were positive in RD1 and nearly 0 in RD2 (Fig. 8a). $Z_m$ was the only parameter which was far from RD2 =0. Of the zooplankton species, *C. affinis* and *O. atlantica* plotted opposite to each other. *O. logicuada*, *C. vanus*, *O. plumifera*, and *F. pellucida* were in the first quadrant, *C. affinis*, *C. sinicus,* and *P. avirostris* in the second, *E. pacifica* and *M. pacifica* in the third, and *O. atlatica* in the fourth. *E. nordmanni*, *M. tenuicornis*, *P. leuckarti*, and *C. pergens* were along RD1 =0, and *P. newmani* was along RD2 =0. Other species were located near the origin. Second, in the combination of RD3 and RD4, the

dominant species, *C. affinis*, *O. atlantica*, *E. nordmanni,* and *P. parvus* s.l., were in the fourth quadrant (Fig. 8b). *O. longicuda* and *C. sinicus* were along RD3 = 0 and RD = 0, respectively. *M. tenuicornis* was in the second quadrant, and *P. avirostris* and *O. plumifera* were in the third (Fig. 8b). On RD3, arrows of SSS and SSChl *a* pointed in opposite directions, and on RD4, bottom depth and $Z_m$ were negatively large (Fig. 8b).

Mean T had the highest coefficient on RD1, 0.043 (Table 2), and those of the other parameters were at least one-fourth that of

mean T; the second highest were 0.009 for SSS and -0.0090 for Year. On RD2, $Z_m$ had the largest absolute coefficient value (-0.47), with mean T (0.027) and bottom depth (0.024) next. On RD3, SSS (0.35) and SSChl *a* (-0.36) were the largest, followed by mean T (-0.020), and on RD4, bottom depth was largest (-0.050), with mean T (-0.027) and $Z_m$ (-0.021) next (Table 2).

In terms of spatial features, RD1 showed a clear west–east trend (Fig. 9a), with significant differences between stations (Fig.

9, ANOVA, $p < 0.01$): RD1 values in Toyama Bay (Stns. 1–8, except 7) were significantly different from those in the western part of the Noto Peninsula to Wakasa Bay (Stns. 16–26) (*Tukey*'s test, $p < 0.001$, Fig. 9d). RD1 was spatially homogenous from Stn. 1 to Stn. 8 except Stn. 7, corresponding to Toyama Bay, and from Stn. 16 to Stn. 26, corresponding to the western part of Noto Peninsula to Wasaka Bay (both areas: *Tukey*'s test, $p > 0.05$). The RD1 of Stn. 7 was significantly different from the other Toyama Bay stations, and similar to Stns. 9–21 except Stns. 17 and 18. The yearly variation of RD1 was smaller

than the spatial variation, based on the range of variation of each station. The RD2–4 did not show west–east trends or significant spatial differences (ANOVA, $p > 0.05$). RD2 was highest at Stn. 8, and most of stations had negative outliers (Fig. 9b). RD3 values were spatially stable (Fig. 9c). The site scores on RD4 were low at Stns. 2, 4, and 8.



## 4. Discussion

In our study, zooplankton abundance differed between the stations (Fig. 6), however, there was no clear west–east gradient. This finding is similar to that of a previous study by Hirota and Hasegawa (1999), which found that zooplankton abundance (wet weight) is homogenous along the Japanese coast at a spatial resolution of $1° \times 1°$.

### 4.1. Interpretation of multivariate analysis

Since the equations with coefficient values calculated from db-RDA are only "hypothesis", the db-RDA results need to be interpreted in terms of the species score. From the db-RDA, variations in zooplankton community structure were mainly distributed along four axes, and most of the variance was explained by RD1. The coefficients (Table 2) show that the variation in RD1 is largely controlled by mean T. Zooplankton species that recorded positive (*C. affinis*, *C. sinicus*, *C. vanus,* and *O. longicauda*) and negative (*O. atlantica*, *E. pacifica,* and *P. newmani*) scores for RD1 (Fig. 8) are classified in the Japan Sea as warm- and cold-water species, respectively (Chiba and Saino, 2003; Iguchi and Tsujimoto, 1997; Iguchi et al., 1999; Takahashi and Hirakawa, 2001). Therefore, this axis is interpreted as explaining spatial differences owing to water temperature, and the equations on RD1 are considered to describe the zooplankton variation well.

RD2, which explained 6.5% of variation, is highly explained by $Z_m$ (Table 2), which is positively related to *C. affinis* and *E. nordmanni* and negatively related to *O. atlantica* (Fig. 8b). The dominance of *C. affinis* and *E. nordmanni* showed high temporal variation (Fig. 6d), and the greatest $Z_m$ was observed in 2007. However, we cannot find a reasonable explanation for the interaction between $Z_m$ and these zooplankton species.

RD3, which explains 4.7% of total variation, is positively affected by SSS and negatively affected by SSChl *a* (Table 2). This relationship has a reasonable explanation: in Toyama Bay, rivers supply nutrients that elevate primary productivity during summer (Terauchi et al., 2014a). Therefore, zooplankton species with negative scores increase in abundance in the eutrophic waters. Large negative scores were observed in the dominant zooplankton species on RD3, except *O. longicauda,* regardless their feeding style: for example, *C. affinis* is carnivorous and *C. sinicus* is a suspension feeder (herbivorous). Appendicularian, including *O. longicauda,* are filter feeders and can prey on nano-plankton size particles (Alldredge and Madin, 1982), which may survive in the oligotrophic conditions.

RD4 explains 3.1% of total variation and is largely affected by bottom depth. The spatial distribution of site scores (Fig. 9) shows such a trend, with high values at Stns. 3 and 5 which are located in the submarine canyon. This axis may reflect the vertical habitat of zooplankton species: *P. avirostris*, which has a largest negative value on RD4, is present >40 m depth, while *E. nordmanni*, which has a positive value, is found at the surface in Toyama Bay (Onbe and Ikeda, 1995). The findings of other studies also support the inference that this axis represents zooplankton vertical habitats. Ueda and Yuasa (2015) found that *C. affinis* is present in the surface layer at the Kuroshio-affected Tosa Bay, and Iguchi (1995) showed that *E. pacifica* is present in deep water (>300 m depth) during daytime in Toyama Bay.

### 4.2. Influence of the CBTWC

RD1 was the only axis that showed differences between the stations and expressed the main feature (84.6%) of spatial variation in the zooplankton community structure in our investigation area. The axis showed greater spatial variation than temporal variation (Fig. 6). RD1 was interpreted as representing the dominance of warm- and cold-water species in Wakasa and



Toyama bays, respectively, which is similar to the findings of previous studies at the same sites (Iguchi and Tsujimoto, 1997; Iguchi et al., 1999). Therefore, this feature is considered highly reproducible with temperature variation in our investigated area in May.

Since RD1 is largely explained by mean T, factors controlling temperature are likely to be responsible for spatial variations in
zooplankton community structure. Three factors potentially control the spatial variation of water temperature: 1) sea surface heat flux; 2) mixing between cold deep-sea water; and 3) heat supply by horizontal advection. The timing of the change from negative to positive sea surface heat flux (Fig. 5a) coincides with that of sea surface temperature increase in Toyama Bay, while it lags that in the other areas (Fig. 3). Additionally, no spatial differences in the sea surface heat flux were observed in the study area (Fig. 5b), which is consistent with previous studies (Hirose et al., 1996; Na et al., 1999). This suggests that sea
surface heat flux is not the controlling factor on spatial variations in temperature except in Toyama Bay, and indicates that other heat supply processes may be more important.

The second potential factor, mixing between cold deep-sea water, initially seems to have greater potential as the bottom depth of stations in Toyama Bay was deeper than elsewhere. Bottom depth was a remaining significant explanatory factor, but its coefficient value was low on RD1 (Table 2). Therefore, mixing between cold deep-sea water is not considered to be the main
factor controlling the spatial variation of temperature, although its influence cannot be completely discounted.

The final potential control on sea surface temperature is horizontal advection by the CBTWC. Previous studies have indicated that the heat content of the Japan Sea is largely affected by horizontal advection associated with the TWC, particularly from April to October (Hirose et al., 1996). In our study, SSH values suggest that the path of the CBTWC was parallel to the mouth of Toyama Bay (Fig. 4), and indicates that the intrusion of the CBTWC is weak and rare in Toyama Bay. Weak and rare
intrusion of the CBTWC in Toyama Bay is supported by previous studies (Hase et al., 1999; Nakada et al., 2002), and is attributed to the break of continental shelf in Toyama Bay (Nakada et al., 2002; Igeta et al., in press). The CBTWC is trapped on the continental shelf and flows along the isodepth from the Tsushima Strait to the Noto Peninsula (Kawabe, 1982); breaks in the continental shelf at Noto Peninsula can potentially alter the direction of the current. The spatial variation of site scores on RD1 corresponds to the bottom topography; higher scores were observed on the continental shelf (Stn. 7 and 9), and there
were lower scores in the downstream reach of the submarine canyon.

Nakada et al. (2002) and Igeta et al. (in press) evaluated the effect of Toyama Bay topography (the submarine canyon, Fig. 1) on the CBTWC using a simple two-layer numerical model. They showed that breaks in the continental shelf induce cyclonic eddies at the edge of the Noto Peninsula, which were sometimes transported to the north, while if the continental shelf is unbroken, the coastal current spreads widely into Toyama Bay. In contrast, the continental shelf is wider at Wakasa Bay (Fig.
1), and water exchange is active during winter (Itoh et al., 2016); here, and along the western part of Noto Peninsula, the zooplankton community structure on RD1 was homogenous. The strong correlation between mean T and the salinity maximum (Fig. 2) also supports that mean T is influenced by the CBTWC.

Based on its influence on temperature variation, the CBTWC can be assumed to be an important control on zooplankton community structure. The importance of ocean currents to zooplankton community structure has also been demonstrated in
coastal areas facing the open ocean, such as the Kuroshio off Japan (Sogawa et al., 2017), and on the Pacific and Atlantic coasts of North America (Mackas et al., 1991; Keister et al., 2011; Pepin et al., 2011; Pepin et al., 2015). In our study, the response of zooplankton to temperature was focused after VIF selection, but the horizontal advection of zooplankton by the CBTWC is also relevant; the importance of the original zooplankton community structure has been demonstrated in a study of





eddies associated with the Leeuwin Current off Australia (Strzelecki et al., 2007). However, in our study, it was difficult to separate the effects of temperature and zooplankton transportation since they were highly correlated.

The spatial variation in Toyama Bay and other areas has been discussed, but differences were also observed between the northern and western parts of the Noto Peninsula, which are downstream and upstream of the CBTWC, respectively (Fig. 4). This difference may results from differences in the original water masses and zooplankton communities. Since the velocity of the CBTWC is ~10 cm s$^{-1}$ and rarely > 30 cm s$^{-1}$ (Hase et al., 1999; Fukudome et al., 2016; Igeta et al., 2011), it takes ~10 days for a water mass to travel 100 km. Wakasa Bay is ~500 km from the Tsushima Strait, so it is possible for water that is present in the Tsushima Strait in March to reach Wakasa Bay and the western part of Noto Peninsula in May; however, it would not yet reach the northern part of Noto Peninsula, which is ~700 km from the Tsushima Strait. Near the Tsushima Strait, dominant copepods in the upstream CBTWC include the warm-water species *C. sinicus*, *P. parvus,* and *C. vanus*, even in March (Hirakawa et al., 1995); cold-water species are present and dominant in the Sea of Japan during winter (Chiba and Saino, 2003).

Pollution (eutrophication) and river discharge are sometimes considered to determine community structure in areas such as the Mediterranean (Siokou-Frangou et al., 1998), the Brazilian coast (Valentin and Monteiroribas, 1993), the coastal area of Taiwan (Chou et al., 2012), and coast of North America (Pepin et al., 2015); however, the low coefficient values of SSChl *a* in our db-RDA results show that eutrophication is not the principal determinant of zooplankton community structure in the Japan Sea based on (Table 2). The cause of this is not clear in this study, but we consider there are two possible explanations: 1) the observation period was at the end of the spring bloom, and local upwelling along the shelf edge in our study area occurs during summer (Nakada and Hirose, 2009); thus the area was not oligotrophic, and the food condition was not important; 2) less-saline water was only present at the surface (< 10 m depth), and most of the water column was not affected by the less-saline eutrophic water.

The dominance of warm-water zooplankton can be treated as the key indicator of the CBTWC. Zooplankton communities have been treated as water mass indicators in other areas (Russell, 1935). Rapid changes in zooplankton community in response to oceanic currents have been identified in areas such as the Kuroshio coast and Mediterranean (Sogawa et al., 2017; Raybaud et al., 2008), but in our study site, the zooplankton community was temporally stable, as indicated by the lower values of Year in RD1 (Table 2). In addition, the dominance of the herbivorous *O. atlantica* in Toyama Bay suggests its food web structure differs from the other areas, which are dominated by the carnivorous *C. affinis* (Ohtsuka and Nishida, 1997); the effect of bottom topography on the path of the CBTWC creates a heterogenic ecosystem along the Japanese coast in the Japan Sea. Previously, it was known that submarine canyons have impacts on the pelagic ecosystem via local upwelling (Fernandez-Arcaya et al., 2017). In Toyama Bay, there is cyclonic circulation when the CBTWC intrudes (Igeta et al., in press), which produces downwelling. Therefore, our results show that changes in the path of the current induced by the submarine canyon promote ecosystem heterogeneity and the rich spatial biodiversity along the coast of Japan.

**Conclusion**

We investigated zooplankton community structure over a 15-year period along the Japanese coast of the Japan Sea, with continental shelf and a submarine canyon. Distance-based RDA indicated that zooplankton community structure is largely





influenced by water temperature of the CBTWC. Warm-water zooplankton were dominant in the path of the CBTWC and along the Japanese coast, and cold-water zooplankton were dominant in Toyama Bay where intrusion of the CBTWC is prevented by a submarine canyon. Therefore, dominance of warm-water species can be used as an index of the CBTWC along the Japanese coast of the Japan Sea. Even though our study area was close to the coast, the effect of land is not dominant, and

biological productivity is mainly controlled by the ocean. Surface waters in the Japan Sea have been affected by global warming and East Asian industrial development for half a century (Belkin, 2009; Kodama et al., 2016). Our study indicates that water temperature largely determines the zooplankton community; therefore, an elevation in sea surface temperature is likely to change zooplankton community structure. Continuous monitoring in our study site helps the effects of global warming on biological productivity to be better understood.

**Acknowledgement**

We thank the captain, officers and crew in the T/V *Mizunagi* of Kyoto Prefecture and R/V *Mizuho-maru* of Japan Fisheries and Education Research Agency cruise for their cooperation at the sea. The daily temperature at eight fixed points were obtained from Japan ocean data center (JODC). The sea surface height and current velocity was obtained from AVISO. The chlorophyll *a* was obtained from GlobColour project. This study was financially supported by general research funds from

Japan Fisheries Research and Education Agency and the Fisheries Agency to all, a research project of the Agriculture, Forestry and Fisheries Research Council (Development of technologies for mitigation and adaptation to climate change in agriculture, forestry and fisheries) to TK, TW and HM, and JSPS grant (16K07831) for TK and TW.

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



**Table 1** Environmental parameters and associated VIF values.

| Parameter | Abbreviation | VIF before selection | VIF after selection |
|---|---|---|---|
| Temperature at 5 m depth | SST | 8.5 | removed |
| Mean temperature in water column | Mean T | 25.9 | 1.8 |
| Minimum temperature in water column | Mini T | 10.9 | removed |
| Salinity at 5 m depth | SSS | 5.4 | 1.5 |
| Mean salinity in water column | Mean S | 6.9 | removed |
| Maximum salinity in water column | Max S | 2.4 | removed |
| Mixed layer depth | $Z_m$ | 1.6 | 1.1 |
| Sea surface chlorophyll *a* concentration | SSChl *a* | 1.1 | 1.0 |
| Bottom depth of station | Bottom depth | 2.0 | 1.4 |
| Investigated year | Year | 1.9 | 1.1 |
| Day of May | Day | 3.2 | removed |
| Observed vertical haul depth of Norpac net | Haul depth | 3.0 | removed |
| Time of day of sampling | Time | 1.1 | 1.0 |





**Table 2** Coefficients for environmental parameters calculated using db-RDA.

|  | RD1 | RD2 | RD3 | RD4 |
|---|---|---|---|---|
| Mean T | 0.043 | 0.028 | -0.020 | -0.027 |
| SSS | 0.009 | -0.003 | 0.035 | -0.002 |
| SSChl $a$ | -0.004 | 0.007 | -0.036 | -0.021 |
| $Z_m$ | 0.001 | -0.047 | -0.008 | -0.007 |
| Bottom depth | -0.005 | 0.025 | 0.013 | -0.052 |
| Year | -0.009 | -0.001 | 0.000 | 0.07 |





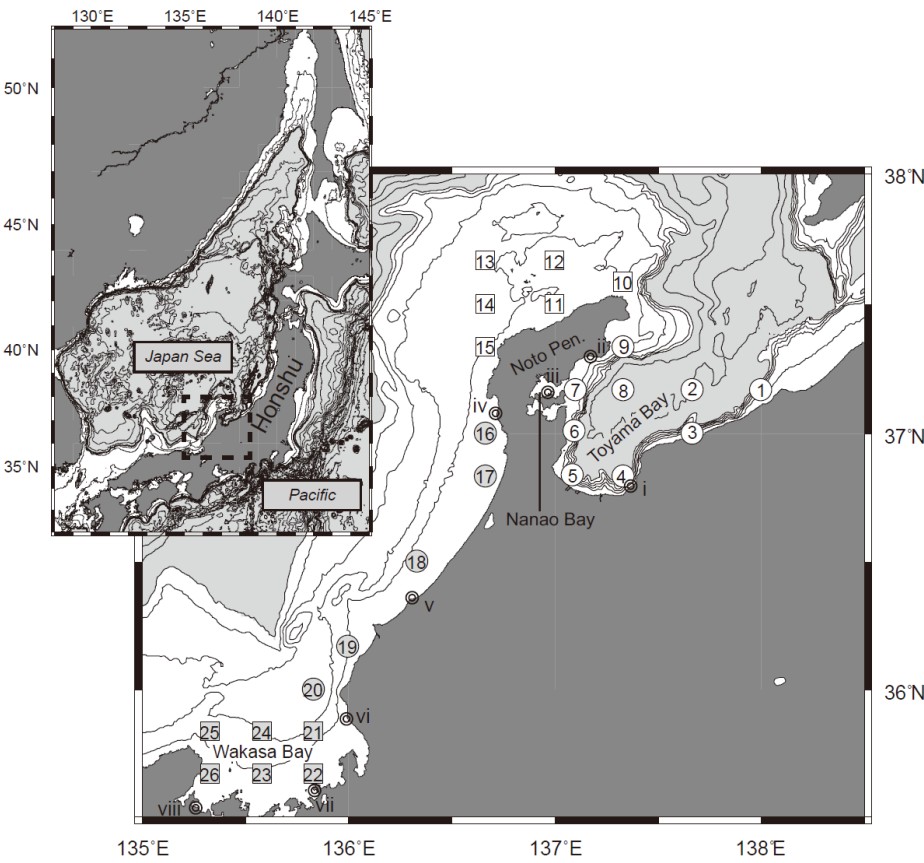

Figure 1: Distribution of sampling stations along the Japanese coast of the Japan Sea. Numbers in open circles, open squares, gray circles and gray squares denote sampling stations in Toyama Bay, northern part of Noto Peninsula, western part of Noto Peninsula, and Wakasa Bay, respectively. Contours indicate bottom depth at 100 m intervals to 500 m depth (white areas),

5 and 500 m intervals thereafter (gray areas). Double circles with small roman numbers (i–viii) denote stations with temperature sensors near the surface. The superimposed map shows location of the study area in the Japan Sea.





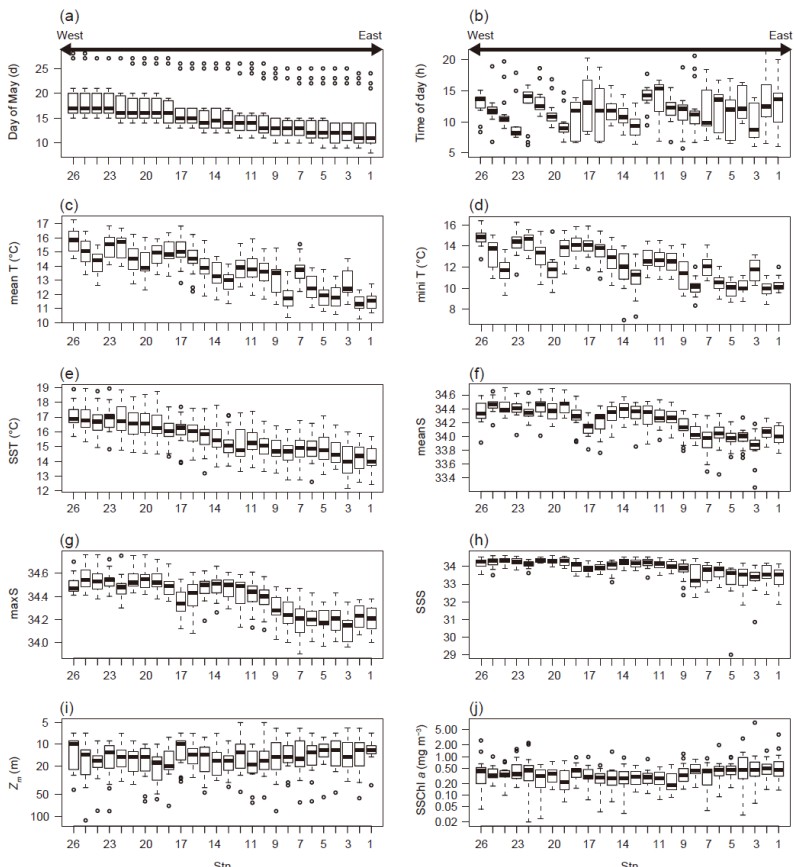

Figure 2: Spatial variations of: (a) day of May; (b) time of day; (c) mean temperature of the water column (mean T); (d) minimum temperature of the water column (mini T); (e) temperature at 5 m depth (SST); (f) mean salinity of the water column (mean S); (g) maximum salinity of the water column (max S); (h) salinity at 5 m depth (SSS); (i) mixed layer depth ($Z_m$); and
5 (j) sea surface chlorophyll *a* concentration in May (SSChl *a*). The X-axis indicates the station number. The plots show the median (horizontal lines within boxes), upper and lower quartiles (boxes), quartile deviation (bars), and outliners (circles).




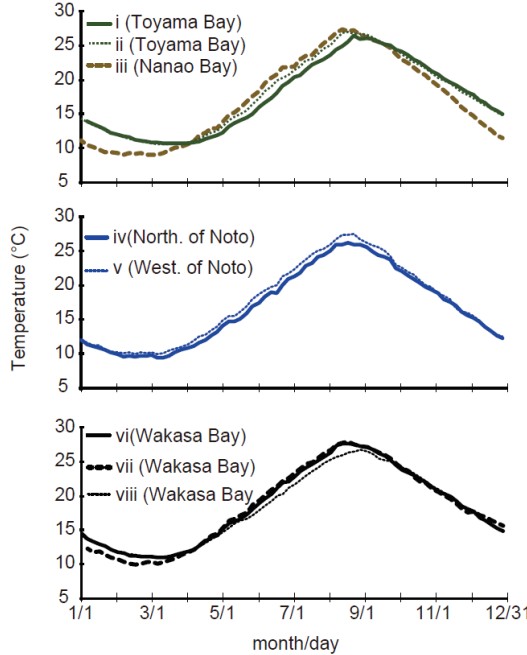

Figure 3: Annual variation in surface temperature at eight fixed stations along the Japanese coast. The upper, middle and bottom panels represent TB (Toyama Bay) and Nanao Bay, NN and WN (northern and western part of Noto Peninsula), and WB (Wakasa Bay), respectively. See Figure 1 for detailed locations.





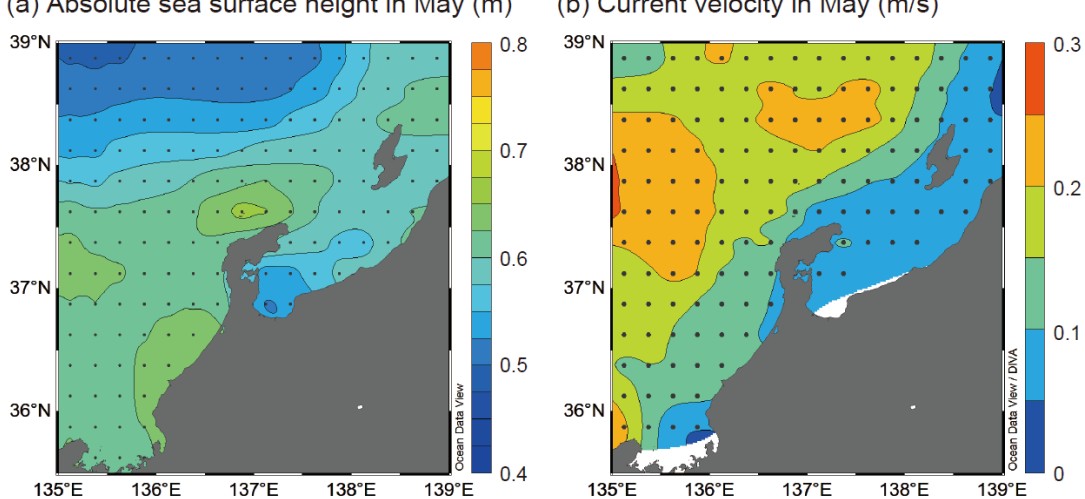

Figure 4: Horizontal distribution of (a) mean absolute sea surface height in May and (b) mean current velocity in May estimated by AVISO.



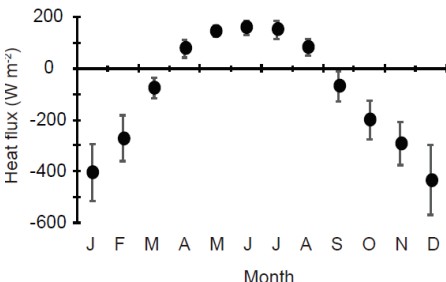

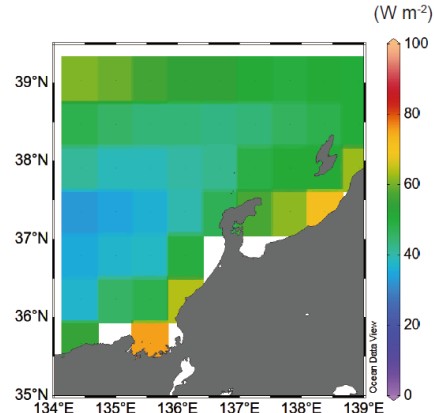

Figure 5: Temporal and spatial variations in sea surface heat flux. (a) Annual variations in sea surface heat flux (W m⁻²) between 2000–2013 in the area of 134–139° E and 35–39°30' N. Vertical bars denote one standard deviation. (b) Spatial variation of sea surface heat flux (W m⁻²) during spring (March–May) from 2000–2013. Blank areas were excluded because they were ascribed as landmass in JRA55.



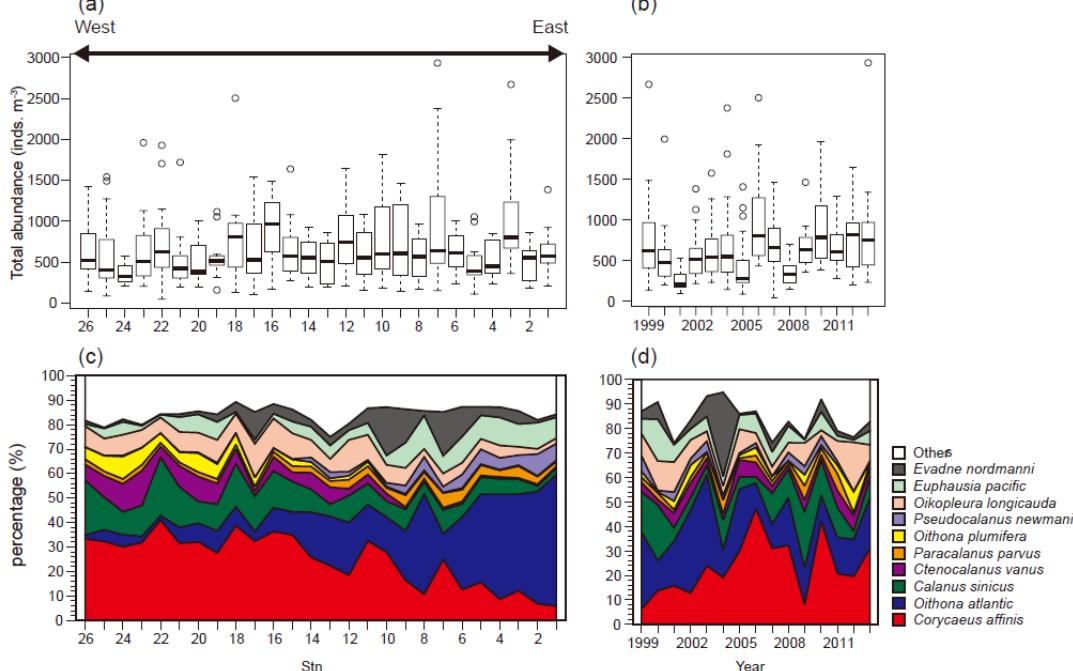

Figure 6: Spatial and yearly variations of total zooplankton abundance (a, b) and contributions of top 10 most abundant species (c, d). The total zooplankton abundance plots show the median (horizontal lines within boxes), upper and lower quartiles (boxes), quartile deviation (bars), and outliners (circles).

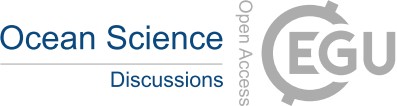

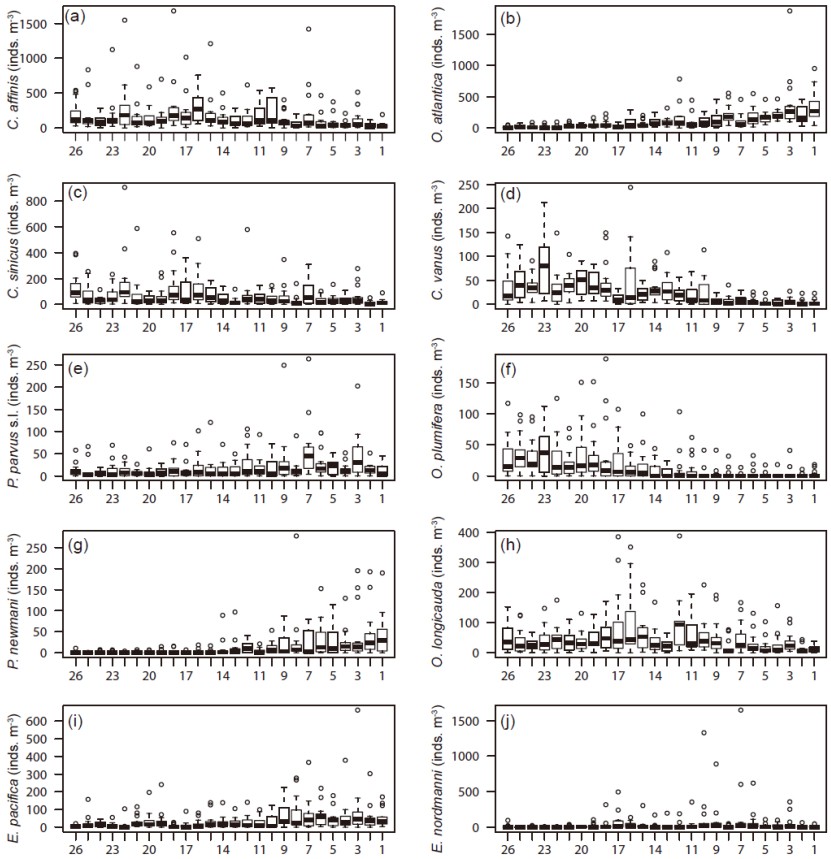

Figure 7: Spatial distribution of the top 10 most abundant zooplankton species, (a) *Corycaeus affinis*, (b) *Oithona atlantica*,
(c) *Calanus sinicus*, (d) *Ctenocalanus vanus*, (e) *Paracalanus parvus* s.l., (f) *Oithona plumifera* (g) *Pseudocalanus newmani*
(h) *Oikopleura longicauda*, (i) *Euphasia pacifica* and (j) *Evadne nordmanni*. The plots in total zooplankton abundance show
median values (horizontal lines within boxes), upper and lower quartiles (boxes), quartile deviation (bars), and outliers (circles).

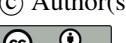


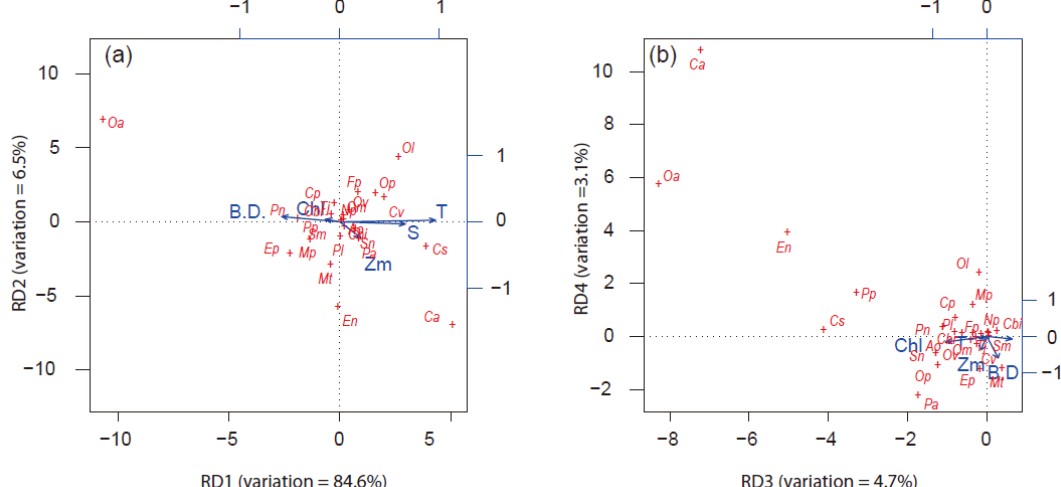

Figure 8: Relationship between environmental parameters (arrows) and zooplankto0n species (crosses) in db-RDA diagram: (a) combination of RD1 and RD2, and (b) combination of RD3 and RD4. The abbreviations T, S, Chl, and B.D. adjacent to arrows represent mean temperature of water column (mean T), salinity at 5 m depth (SSS), sea surface chlorophyll *a* (SSChl *a*), and bottom depth of station, respectively. The 25 zooplankton spices were represented by their initials. The dashed lines denote the axis = 0.


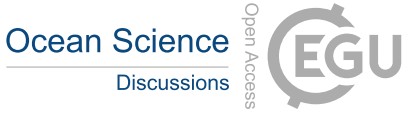

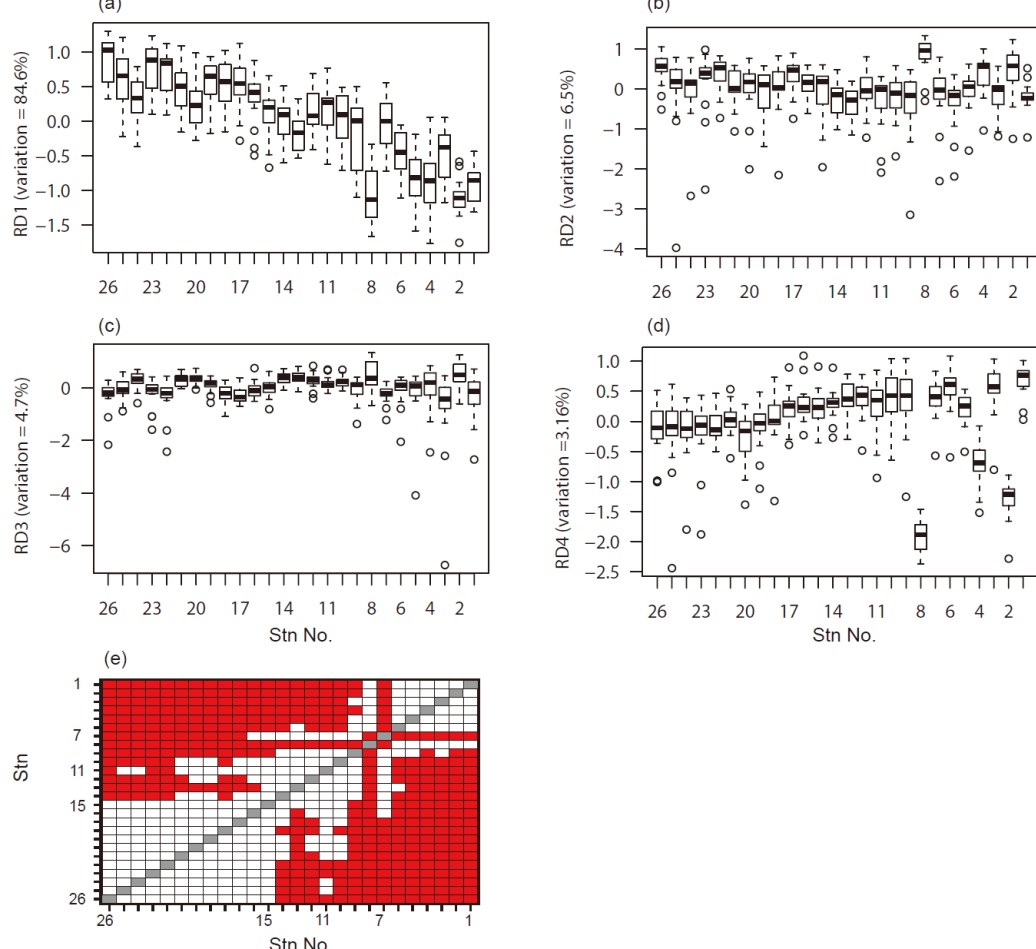

Figure 9: Spatial variation of station db-RDA scores (left panels): (a) RD1, (b) RD2, (c) RD3 and (d) RD4. Plots show the median (horizontal lines within boxes), upper and lower quartiles (boxes), quartile deviation (bars), and outliers (circles). (e) The results of Tukey's HSD test between stations (right panels) onRD1. Red tiles indicate significant differences ($p < 0.05$).