# Peer review of "Spatial variations in zooplankton community structure along the Japanese coastline in the Japan Sea: influence of the coastal current Taketoshi Kodama1,\*, Taku Wagawa1, Naoki Iguchi1, Yoshitake Takada1, Takashi Takahashi2, Ken-Ichi Fukudome<sup"

_Ocean Science, 2017_

## Referee Comment (RC1) · Anonymous Referee #1 · 2 Jan 2018

Spatial variations in zooplankton community structure along the Japanese coastline in the Japan Sea: influence of the coastal current

General comments:

This paper presents results from a time series on zooplankton community composition and for concomitant environmental data collected in the month of May from 1999-2013 along the Japanese coastline. Biological and physical data have been collected and analyzed with timely methods and hold the potential to increase the understanding of the factors controlling zooplankton community composition on the mesoscale and the physical/environmental factors affecting this composition in the decadal time frame.

[Figure]

However, the graphical presentation of results as well as the discussion of findings need to be revised. Especially, the evidence presented to support the effect of the coastal current is not convincing in its present form. I therefore suggest a major revision of the manuscript before possible publication.

Specific comments:

Introduction

Page 1, line 29/30: Effects of currents on zooplankton community composition are the core of this manuscript and the existing literature on this topic is extensive. The authors should review the publications from study areas with similar environmental conditions. Also, other factors that drive community composition on the regional scale should be addressed, e.g. seasonality or nutrient loading.

Material and Methods

Page 3, line 7: For a more complete assessment of zooplankton community structure, night time samples are preferably used, since they also take into account the migrating part of the mesozooplankton community. However, tows from 150m to the surface included the water column well below the mixed layer. I therefore think that the sampling was appropriate to evaluate mesozooplankton community composition.

Page 3, lines 28/29: Give the reference for equation (1)

Page 4, line 5: Although the discarding of rare zooplankton species for statistical reasons is a necessary step, this leads to a loss of ecological knowledge. Rare species are indicative of particular environmental conditions or events, e.g. the advection of water masses. One way of keeping this information in the analysis is the use of species richness (S) as a variable.

Page 4, line 13-20: It is not clear on which reasoning the choice of parameters is based. Parameters with VIF values between 3 and 10 are discarded, but mean temperature with a very high VIF value (25.9!) is kept in the analysis and reveals itself as a parameter with high explanatory power (e.g. page 7, line 9 "the variation in RD1 is largely controlled by mean temperature"). The authors should state more clearly their procedure in calculating VIF values before and after selection, and why mean temperature is preferred to temperature at 5m depth, for example.

Figure 1: Present a map where the position of the current is indicated.

Results

Page 4, line 34f and Figure 2: spatial trends mini T, SST, max S are not used further in the analysis and discussion. I suggest to present only panels a, b, c, h, i, and j in Figure 2.

Page 5, line 3: "monthly SSChla concentration was variable"; from Figure 2j is appears that monthly SSChla is in fact very low and stable in spatial and temporal terms.

Page 5, line14: "78 of the 388 samples were identified to species level and 25 groups to genus level"; From this sentence is it not clear whether 310 samples of the 15-year time series were not analyzed at all, and if yes, how were the 78 samples chosen? What do you mean by groups? I presume it is taxonomic groups. How were these groups used in the statistical analysis? Add this information to the Material and Methods section.

General comment on figures: The manuscript contains 9 Figures with numerous panels. Several figures present information that is redundant and some information is not used in the discussion. I suggest substantial revision of the figures and focus on the relevant information.

Figure 3: The surface temperature is presented, but the mean temperature is used for statistical analysis and for the discussion of results. I do not think this figure is necessary to be presented.

Figure 4: indicate the position of the coastal current in panel 4b

Figure 5: plot the current vectors in panel 5b to indicated direction of flow and current

speed

Figure 6: revise the x-axis in panels b) and d) so that the axis scaling and years are aligned

Figure 7: the information presented here is in part redundant with information presented in Figure 6c

Figure 9: make a separate figure for panel 9e or move to supplementary material

Discussion:

My major concern with this publication is the discussion. Overall, the discussion needs to be re-structered.

The information obtained from the zooplankton community analysis is not properly discussed in the light of ecological differences between the sampling sites. See the publication by Espinasse et al. 2014, Mar Ecol Prog Ser. Vol. 506: 31–46; doi: 10.3354/meps10803 as just one example.

From your data, it is clear that Toyama Bay has a very different zooplankton community structure compared to the stations along the coast. However, it is not clear whether this region is influenced by the coastal current or not (see page 8 line 19f and line 26ff). The possible role of nutrient input or bottom topography is only marginally addressed in this manuscript and needs to be elaborated.

The occurrence of key organisms such as Oithona atlantica needs to be discussed. Turbulent motion is possibly one of the factors that contribute to its spatial distribution. See for example the paper by Saiz et al. 2003 (Limnol Oceanogr Volume 48, Issue 3, Pages 1304–1311)

Also, the evidence presented for the role of the coastal current is not convincing. The most relevant parameter (mean temperature) has the highest VIF (25.9, before selection) and caution should be given when using it in the statistical analysis. However,

the spatial variation of RD1 and its explanatory power (84 %) rely on mean temperature. Mean temperature is of high biological relevance, since it affects all metabolic processes (feeding, growth, reproduction) in the zooplankton. To show an effect of oceanographic parameters (i.e. currents) the use of salinity and temperature at a certain depth is possibly more appropriate. I suggest to repeat the statistical analysis using S and T at 5 m depth and to compare the results with your findings when you use mean T.

Sea surface heat flux is discarded as a factor influencing spatial variations (i.e. the east-west trend), and, ultimately, the occurrence of warm water and cold water species (page 8, lines 4-11). In a recent publication, Smyth et al. 2014 (PLOS one, Volume 9 | Issue 6 | e98709) use sea surface heat flux as a forcing factor in the seasonal structure of the pelagic ecosystem. The authors should re-discuss their findings in the light of these observations.

---

## Referee Comment (RC2) · Anonymous Referee #2 · 23 Feb 2018

The manuscritp has interesting and important information of the composition of the zooplankton communities, their changes during 15 years, and their relation with enviromental data. Therefore, it presents sustancial data that contribute to the knowledge of the complex zooplankton communities.

Specific comments: - It is neccesary to do a careful review of the construction of the lenguaje because few sentences are incompletes since some verbs or connectors words are absent.

- Page 2, lines 1-7 review the chronology order of the citations of some authors.

- Page 2, line 14 the generic and specific names have typing errors, they must write

[Figure]

Paracalanus parvus not Palacalanus pavrus.

- In my opinion should be more useful to consider the temperature in the water columm (if the data were obtained with the STD) that the water column mean.

- Several data, as the surface temperature obtained at fixed points are not used in the analysis and discussion, so the question is, if these data contribute or not with relevant information in the studied period. If not, it is posible to reduce data that could help to clarified the text.

---

## Author Comment (AC2) · 22 Mar 2018

**Response to the specific comments of R#2**

It is neccesary to do a careful review of the construction of the lenguaje because few sentences are incompletes since some verbs or connectors words are absent.

**We appreciate to this comment: we re-checked the text.**

Page 2, lines 1-7 review the chronology order of the citations of some authors.

**We checked the published one and revised as you suggested (P2 line 5-9).**

Page 2, line 14 the generic and specific names have typing errors, they must write Paracalanus parvus not Palacalanus pavrus.

**We are sorry for careless mistake. We revised (P2, line 19).**

In my opinion should be more useful to consider the temperature in the water columm (if the data were obtained with the STD) that the water column mean.

**We agreed with your comments; using the water temperature of fixed depth is more informative, however, the bottom depth ranged largely (40-1200 m), and thus we used ~30 m depth of temperature in the water column. When some data was not available (NA), we must remove the station which contained NA value from the db-RDA. It decreases the number of samples, and thus we adopt the mean temperature in this study.**

Several data, as the surface temperature obtained at fixed points are not used in the analysis and discussion, so the question is, if these data contribute or not with relevant information in the studied period. If not, it is posible to reduce data that could help to clarified the text.

**We removed the description and this figure in the main text.**

[revised manuscript text omitted]

---

## Author Response (AR1)

We appreciate your careful and insightful comments on our paper. We consider our revised manuscript is significantly improved based on your comments. The revised parts were marked with yellow marker.

By the way, we apologize that we have find inappropriate part in our R program: we should treat "year" as categorical values to remove in the partial db-RDA analysis, but we had treated as numeric values in the previous ms. This means that the yearly variation had been removed from the partial db-RDA analysis by linear function; we considered that the yearly variation is not always linear, and thus revised in this ms. In addition, Day of May were also removed from the analysis to focus on the spatial variation. As the results of this revision, the major variation (RD1) was unchanged, but the minor variation (RD2-4) was significantly changed. The results and discussion (session 4.1) were revised with this revision. We considered this revision make our message clearer: the spatial variation more clearly shown in the revised ms. The revised parts were marked with blue marker concerning to this revision.

General comments:
However, the graphical presentation of results as well as the discussion of findings need to be revised. Especially, the evidence presented to support the effect of the coastal current is not convincing in its present form.

We deeply appreciate your many valuable comments on our ms; we revised based on your comments and we feel our ms is improved very much. In particular, we revised figures and discussions. The specific revised points were described corresponding to the specific comments.

Specific comments: Introduction

1. Page 1, line 29/30: Effects of currents on zooplankton community composition are the core of this manuscript and the existing literature on this topic is extensive. The authors should review the publications from study areas with similar environmental conditions. Also, other factors that drive community composition on the regional scale should be addressed, e.g. seasonality or nutrient loading.

**We thank this comment. We added the reviews of zooplankton variation in the first paragraph: the studies based on the long-term monitoring in the English Channel were added. In addition, we also added that the zooplankton community structure was not only decided by ocean currents.**

Material and Methods

2. Page 3, line 7: For a more complete assessment of zooplankton community structure, night time samples are preferably used, since they also take into account the migrating part of the mesozooplankton community. However, tows from 150m to the surface included the water column well below the mixed layer. I therefore think that the sampling was appropriate to evaluate mesozooplankton community composition.

**We appreciate this comment. We consider the vertical migration is usually limited because of the shallow bottom depth in many stations, while at some stations in Toyama Bay, it should be considered because diel vertical migration of *Euphausia pacifica* and *Metridia pacifica* are reported. We added the discussion about this (at 4.1. P7 line 2-10)**

Page 3, lines 28/29: Give the reference for equation (1)

**We used the equation shown in the database (JRA-55). We added the web address of the JRA-55 atlas (P3 line 34).**

Page 4, line 5: Although the discarding of rare zooplankton species for statistical reasons is a necessary step, this leads to a loss of ecological knowledge. Rare species are indicative of particular environmental conditions or events, e.g. the advection of water masses. One way of keeping this information in the analysis is the use of species richness (S) as a variable.

**We calculated the species richness and described in the results session. It was significantly different among the stations, but not observed the west-east trend (P3 line 29-30).**

Page 4, line 13-20: It is not clear on which reasoning the choice of parameters is based. Parameters with VIF values between 3 and 10 are discarded, but mean temperature with a very high VIF value (25.9!) is kept in the analysis and reveals itself as a parameter with high explanatory power (e.g. page 7, line 9 "the variation in RD1 is largely controlled by mean temperature"). The authors should state more clearly their procedure in calculating VIF values before and after selection, and why mean temperature is preferred to temperature at 5m depth, for example.

**We are sorry for the inconvenient text in the previous manuscript. We added the more detail explanations. When we remove the parameters with highest VIF, the temperatures (SST, minimum T, and mean T) were removed firstly, and cannot use them in the db-RDA analysis. Temperature is distinctly an important parameter which explain the zooplankton variations; thus we want to use at least one of them. Additionally, we considered that the mean-temperature is the representative**

parameters of these parameters: the highest VIF value in these parameters were the results of the correlation of the other factors. Thus, we keep this parameter, and removed the other parameters with >3 VIF. In addition to this explanation, we checked that the other parameters were possible to explain the variation, as followed the other comments for discussion the importance of Tsushima Warm Current (P4 line 24-29).

Figure 1: Present a map where the position of the current is indicated.

We added the flows according to Hase et al 1999 J Oceanogr (Fig. 1, P17).

Results
Page 4, line 34f and Figure 2: spatial trends mini T, SST, max S are not used further in the analysis and discussion. I suggest to present only panels a, b, c, h, i, and j in Figure 2.

We removed from these panels, but we keep max and SST because they were used in the discussion (Fig. 2, P18).

Page 5, line 3: "monthly SSChla concentration was variable"; from Figure 2j is appears that monthly SSChla is in fact very low and stable in spatial and temporal terms.

We revised the descriptions (P5, line 10-13).

Page 5, line14: "78 of the 388 samples were identified to species level and 25 groups to genus level"; From this sentence is it not clear whether 310 samples of the 15-year time series were not analyzed at all, and if

yes, how were the 78 samples chosen? What do you mean by groups? I presume it is taxonomic groups. How were these groups used in the statistical analysis? Add this information to the Material and Methods section.

**We are sorry for accomplished description. We revised (P5, line 18-19).**

General comment on figures: The manuscript contains 9 Figures with numerous panels. Several figures present information that is redundant and some information is not used in the discussion. I suggest substantial revision of the figures and focus on the relevant information.

**We removed the unnecessary figures from the revised manuscript and limited to 7 Figs. The details were described below.**

Figure 3: The surface temperature is presented, but the mean temperature is used for statistical analysis and for the discussion of results. I do not think this figure is necessary to be presented.

**We removed this fig as well as the descriptions: the data was not used in the discussion.**

Figure 4: indicate the position of the coastal current in panel 4b
Figure 5: plot the current vectors in panel 5b to indicated direction of flow and current speed

**We consider both of these comments are for Figure 4. We added the current vector in this figure, and revised as Fig. 3 (P19).**

Figure 6: revise the x-axis in panels b) and d) so that the axis scaling and years are aligned

**We revised this figure as Fig 5 (P21).**

Figure 7: the information presented here is in part redundant with information presented in Figure 6c

**We deleted this figure in the revised ms.**

Figure 9: make a separate figure for panel 9e or move to supplementary material

**We deleted this panel from the figure (Fig. 7, P23).**

Discussion:
My major concern with this publication is the discussion. Overall, the discussion needs to be re-structered.

**We revised the discussion part based on your comments as follows.**

The information obtained from the zooplankton community analysis is not properly discussed in the light of ecological differences between the sampling sites. See the publication by Espinasse et al. 2014, Mar Ecol Prog Ser. Vol. 506: 31–46; doi: 10.3354/meps10803 as just one example.

From your data, it is clear that Toyama Bay has a very different

zooplankton community structure compared to the stations along the coast. However, it is not clear whether this region is influenced by the coastal current or not (see page 8 line 19f and line 26ff). The possible role of nutrient input or bottom topography is only marginally addressed in this manuscript and needs to be elaborated.

The occurrence of key organisms such as Oithona atlantica needs to be discussed. Turbulent motion is possibly one of the factors that contribute to its spatial distribution. See for example the paper by Saiz et al. 2003 (Limnol Oceanogr Volume 48, Issue 3, Pages 1304–1311)

**Following to these three comments, we added the discussion on the uniqueness of Toyama Bay in the view of bottom depth (submarine canyon structure), and eutrophication (P8, line 3-29). The bottom depth must be important, because the deep layer is the habitat of cold water species. However, we considered that the diel migrators are not majorly affected to zooplankton community in our study, because our observation was mainly conducted in daytime. The importance of eutrophication with river discharge was unclear in this study; it may be affected to the dominance of herbivore *Oithona* in Toyama Bay, however, we cannot show the evidence. In addition, the development of stratification with less-saline river discharge may contribute the dominance of *Oithona*.**

Also, the evidence presented for the role of the coastal current is not convincing. The most relevant parameter (mean temperature) has the highest VIF (25.9, before selection) and caution should be given when using it in the statistical analysis. However, the spatial variation of RD1 and its explanatory power (84 %) rely on mean temperature. Mean temperature is of high biological relevance, since it affects all metabolic processes (feeding, growth, reproduction) in the zooplankton. To show an effect of oceanographic parameters (i.e. currents) the use of salinity and temperature at a certain depth is possibly more appropriate. I suggest

to repeat the statistical analysis using S and T at 5 m depth and to compare the results with your findings when you use mean T.

**We did the statistical analysis using max S and SST instead of SSS and mean T (P4, line 37-P5, line 1). The temperature and salinity at the certain depth is considered as the good indicator, however, it was difficult to adopt in this study because observations were only conducted ~40 m depth at some of stations. As the results, the coefficients of SST and maximum salinity were 0.0392 and 0.0192, respectively, which were highest and second highest in the equation. The spatial variation of SST was considered the results of the net heat flux, water temperature in the last winter, and advection of coastal branch of Tsushima Warm Current, whereas max S is the indicator of the Tsushima Warm Current itself, because the maximum salinity is increased tendency at the Tsushima Strait, origin of the TWC. Base on this result, we consider that at least one-third of variation is explained by the advection of TWC. We added this discussion to P9 line 30-36.**

Sea surface heat flux is discarded as a factor influencing spatial variations (i.e. the east-west trend), and, ultimately, the occurrence of warm water and cold water species (page 8, lines 4-11). In a recent publication, Smyth et al. 2014 (PLOS one, Volume 9 | Issue 6 | e98709) use sea surface heat flux as a forcing factor in the seasonal structure of the pelagic ecosystem. The authors should re-discuss their findings in the light of these observations.

**NHF is an important factor, for example, it controls the onset of spring bloom. However, in present study, we cannot find the spatial variations, and thus NHF cannot explain the spatial variation of zooplankton community, as we described in the previous ms. In Smyth et al. 2014, increase of zooplankton abundance is not matched with NHF; we considered that zooplankton community structure was not explained "on-site" environmental variation; the teleconnection is important via the ocean current based on our results (P9, line 5-8).**

[revised manuscript text omitted]

---

## Author Response (AR2)

Response to Editor

We appreciate the careful review and comments to our manuscript. We revised our manuscript based on all of the comments.

> *You are using data from other sources, e.g. GlobColour and AVISO. Please check the fair data use statements of those data providers whether the acknowledgement is correct.*

We revised as following the statements of those data providers (P11, L3–6).

> *I propose to use less abbreviations, like min S, or max T.*

We changed without use of abbreviations to minimum temperature, mean salinity, and maximum salinity, which were not frequently used in the manuscript (P15, Table 1).

- Specific comments

  > *P1, L30-35 Please change to: Ocean currents are important for zooplankton abundance and community structure: the Gulf Stream affects the abundance of copepods, Calanus finmarchicus and C. helgolandicus, in the European coastal seas (Frid and Huliselan, 1996; Reid et al., 2003), and the copepod species richness varies with the inflow of source water in the California current (Hooff and Peterson, 2006). Not only ocean currents, but also water temperature, nutrient supply, and many other factors affect zooplankton abundance and community structure; thus, monitoring results of zooplankton abundance and community usually reflect hydro-meteorological change (Beaugrand, 2005).*

We revised as suggested (P1, L30–35)

> *P2, L4-5 change to: The TWC transport affects climate and marine systems in*

*the Japan Sea and the Japanese Archipelago.*

We revised as suggested (P2, L4).

*P2, L17-18 change to: … but in two studies (Iguchi and Tsujimoto, 1997; Iguchi et al., 1999) heterogeneity in zooplankton community structure during spring bloom period was identified:*

We revised as suggested (P2, L15–17).

*P3, L6 discarded (not: omitted)*

We revised as suggested (P3, L4).

*P3, L7 Change to: As the temperature- and density-based definition of the depth of the mixed layer …*

We revised as suggested (P3, L5).

*P3, L8 Japan (typo)*
*P3, L13 (from 6 to 19 hrs)*

We revised as suggested (P3, L6 and L11).

*Section 2.1 Please provide precision and accuracy of the measurements.*

We considered that it is difficult for quantitative evaluations of precision and accuracy in zooplankton identification. Understanding our quality of zooplankton data set for readers,

we added following sentences concerning the detailed methods for identification (P3, L13–21): "In the laboratory, all large zooplankton (ca. >2 mm) in a sample were picked up for identification, and then the remining sample was divided into subsamples using a Folsom or Motoda plankton splitter or a wide-bore pipette.   The subsamples were split as the copepods in each subsample were 150–200 individuals.   Individuals were identified to species level based on Chihara and Murano (1997), and then the density of each zooplankton species was calculated (individuals per cubic meter, inds. m$^{-3}$).   In case of copepods, only adults and C4 and C5 stages of copepodites were identified and counted.   The subsamples were observed until the number of the dominant zooplankton species exceeded 150 inds; 2% of a sample were observed in average.   The people identified zooplankton species were same in our study period; this means constant levels of accuracy and precision in the zooplankton community identification were maintained."

*P4, L27 remaining (not: remained)*
*P5, L6 clearly lower at (not: lowered to)*
*P5, L9 discharge (not: flow)*
*P7, L5, L6 diel (typo)*

We revised as suggested (P4, L29; P5, L9; P5, L12; P7 L5, 6).

*P7, L7-10 Change to (if my interpretation is correct): Our consideration for rejecting the sampling time from the explained values was that the sampling was mostly done in the day-time (Fig. 2b), and the bottom depth is shallow in the western part of our sampling site (Fig. 1). On the other hand, our results show the day-time zooplankton community structure: the variation of night-time may be different.*

We revised as suggested (P7, L7–10).

*P7, L18-19 Change to (if my interpretation is correct): Although RD2 and RD3*

*only show a minor contribution to the spatial variation of zooplankton community structure, we attempted to interpret the data.*

We revised as suggested (P7, L18-19).

*P7, L23 increases*

We revised as suggested (P7, L23).

*P7, L28-30 Please modify this sentence, because it is not clear what is meant.*

We revised as "The resting eggs of *E. nordmanni* which only sink to the shallow sea bottom will hatch (Onbé, 1985); the shallow bottom depth is important for its reproduction, and thus the abundance was considered to be high in the shallow bottom areas." (P7, L28–30).

*P8, L5 I think this should be: These affect the zooplankton community as well as temperature.*

We revised as suggested (P8, L5).

*P8, L7-8 Change to: For example, in the Mediterranean Sea the river input changes the amount of phytoplankton, which in turn affects the amount of zooplankton (Espinasse et al., 2014).*

We revised as suggested (P8, L7–8).

*P8, L11-12 "The results of RD1 were contained the bottom depth in the*

*regression analysis, but the coefficient of bottom depth was significantly lower than the mean T (Table 2).” I do not understand this sentence exactly, please change it.*

*P8, L12-14 Change this sentence to: This indicated that the bottom depth did not directly contribute to the zooplankton community structure, although in other studies it was shown that the deep layer is the habitat of zooplankton which migrate to shallower depth (e.g. Herman et al., 1991).*

We revised these sentences (P8, L11-15).

*P8, L30 delete: As the results*
*P8, L32 with (not: between)*

We revised these parts as suggested (P8, L31 and L33).

*P9, L5-7 Change to: We consider that the zooplankton abundance and community structure is generally not explained by “on-site” (in situ) environmental variations: large ocean systems can also influence the local zooplankton community variations, which is also observed in European seas.*

We revised these parts as suggested (P9 L6–8).

*P10, L12 I do not understand the use of reproducible here. Please explain and change.*

We revised the sentence as “, which were different in our study site; similar zooplankton community was observed every year in May.” (P10, L12).

*Ohtsuka and Nishida: Japanese (typo)*

We revised (P13, L27; L33).

*Figure 1 Please indicate what the arrows exactly represent.*

We added (P17).

*Figure 3 (counters) should be (colors), right?*

Yes, we revised (P19).

[revised manuscript text omitted]
. In the laboratory, all large zooplankton (ca. >2 mm) in a sample were picked up for identification, and then the remining sample was divided into subsamples using a Folsom or Motoda plankton splitter or a wide-bore pipette. The subsamples were split as the copepods in each subsample were 150–200 individuals. Individuals were identified to species level based on Chihara and Murano (1997), and then the density of each zooplankton species was calculated (individuals per cubic meter, inds. m$^{-3}$). In case of copepods, only adults and C4 and C5 stages of copepodites were identified and counted. The subsamples were observed until the number of the most dominant copepod species exceeded 150 inds; 2% of a sample were observed in average. The people identified zooplankton species were same in our study period; this means constant levels of accuracy and precision in the zooplankton community identification were maintained. 
[revised manuscript text omitted]

The importance of ocean currents to zooplankton community structure has also been demonstrated in coastal areas facing the open ocean, such as the Kuroshio off Japan (Sogawa et al., 2017), and on the Pacific and Atlantic coasts of North America (Mackas et al., 1991; Keister et al., 2011; Pepin et al., 2011; Pepin et al., 2015). In our study, the response of zooplankton to temperature was focused after VIF selection, but the horizontal advection of zooplankton by the CBTWC is also relevant; the importance of the original zooplankton community structure has been demonstrated in a study of eddies associated with the Leeuwin Current off Australia (Strzelecki et al., 2007). When the other db-RDA was done with max salinity and SST as the explanatory parameters instead of mean T and SSS, the coefficients of SST and maximum salinity were 0.0392 and 0.0192, respectively. The SST variation was considered as the results of the CBTWC, water temperature in winter and Qnet, while the maximum salinity was one of the index of CBTWC transport, because the maximum of salinity of the TWC is increased

from spring to summer at the Tsushima Strait (Morimoto et al., 2009). Therefore, we considered that the contributions of CBTWC was at least one-third of RD1 variation. Hence, as it was difficult to completely separate the effects of temperature and zooplankton transportation, the CBTWC can be assumed to be an important control on zooplankton community structure. The spatial variation in Toyama Bay and other areas has been discussed, but differences were also observed between the northern and western parts of the Noto Peninsula, which are downstream and upstream of the CBTWC, respectively (Fig. 4). This difference may result from differences in the original water masses and zooplankton communities. Since the velocity of the CBTWC is ~10 cm s$^{-1}$ and rarely > 30 cm s$^{-1}$ (Hase et al., 1999; Igeta et al., 2011; Fukudome et al., 2016), it takes ~10 days for a water mass to travel 100 km. Wakasa Bay is ~500 km from the Tsushima Strait, so it is possible for water that is present in the Tsushima Strait in March to reach Wakasa Bay and the western part of Noto Peninsula in May; however, it would not yet reach the northern part of Noto Peninsula, which is ~700 km from the Tsushima Strait. Near the Tsushima Strait, dominant copepods in the upstream CBTWC include the warm-water species *C. sinicus*, *P. parvus,* and *C. vanus*, even in March (Hirakawa et al., 1995); cold-water species are present and dominant in the Sea of Japan during winter (Chiba and Saino, 2003).

The dominance of warm-water zooplankton can be treated as the key indicator of the CBTWC. Zooplankton communities have been treated as water mass indicators in other areas (Russell, 1935). Rapid changes in zooplankton community in response to oceanic currents have been identified in areas such as the Kuroshio coast and Mediterranean (Raybaud et al., 2008; Sogawa et al., 2017), which were different in our study site; similar zooplankton community was observed every year in May. In addition, the dominance of the herbivorous *O. atlantica* in Toyama Bay suggests its food web structure differs from the other areas, which are dominated by the carnivorous *C. affinis* (Ohtsuka and Nishida, 1997); the effect of bottom topography on the path of the CBTWC creates a heterogenic ecosystem along the Japanese coast in the Japan Sea. Previously, it was known that submarine canyons have impacts on the pelagic ecosystem via local upwelling (Fernandez-Arcaya et al., 2017). In Toyama Bay, there is cyclonic circulation when the CBTWC intrudes (Igeta et al., 2017), which produces downwelling. Therefore, our results show that changes in the path of the current induced by the submarine canyon promote ecosystem heterogeneity and the rich spatial biodiversity along the coast of Japan.

**5. Conclusion**

We investigated zooplankton community structure over a 15-year period along the Japanese coast of the Japan Sea, with continental shelf and a submarine canyon. Distance-based RDA indicated that zooplankton community structure is largely influenced by water temperature of the CBTWC. Warm-water zooplankton were dominant in the path of the CBTWC and along the Japanese coast, and cold-water zooplankton were dominant in Toyama Bay where intrusion of the CBTWC is prevented by a submarine canyon. Therefore, dominance of warm-water species can be used as an index of the CBTWC along the Japanese coast of the Japan Sea. Even though our study area was close to the coast, the effect of land is not dominant, and biological productivity is mainly controlled by the ocean. Surface waters in the Japan Sea have been affected by global warming and East Asian industrial development for half a century (Belkin, 2009; Kodama et al., 2016). Our study indicates that water temperature largely determines the zooplankton community; therefore, an elevation in sea surface temperature is

likely to change zooplankton community structure.  Continuous monitoring in our study site helps the effects of global warming on biological productivity to be better understood.

**Acknowledgement**

We thank the captain, officers and crew in the T/V *Mizunagi* of Kyoto Prefecture and R/V *Mizuho-maru* of Japan Fisheries
5   and Education Research Agency cruise for their cooperation at the sea.  ==The altimeter products were produced by Ssalto/Duacs and distributed by Aviso, with support from Cnes (http://www.aviso.altimetry.fr/duacs/)==.  The chlorophyll *a* was obtained from GlobColour project: ==GlobColour data (http://globcolour.info) used in this study has been developed, validated, and distributed by ACRI-ST, France==.  We deeply appropriate an anonymous referee, and Dr. Sabine Schultes for their insightful comments in our discussion paper.  
[revised manuscript text omitted]